# Semantic Segmentation Dataset for AI-Based Quantification of Clean Mucosa in Capsule Endoscopy

**DOI:** 10.3390/medicina58030397

**Published:** 2022-03-07

**Authors:** Jeong-Woo Ju, Heechul Jung, Yeoun Joo Lee, Sang-Wook Mun, Jong-Hyuck Lee

**Affiliations:** 1Biomedical Research Institute, Pusan National University Yangsan Hospital, Yangsan 50612, Korea; veryju@gmail.com (J.-W.J.); soulwalker@naver.com (S.-W.M.); 2Captos Co., Ltd., Yangsan 50652, Korea; 3Department of Artificial Intelligence, Kyungpook National University, Daegu 41566, Korea; heechul@knu.ac.kr; 4Department of Pediatrics, Pusan National University School of Medicine, Pusan National University Yangsan Hospital, Yangsan 50612, Korea; 5Seoreu Co., Ltd., Busan 46288, Korea; jhlee@seoreu.com

**Keywords:** semantic segmentation, deep learning, capsule endoscopy, small bowel cleanliness, visualization scale

## Abstract

*Background and Objectives*: Capsule endoscopy (CE) for bowel cleanliness evaluation primarily depends on subjective methods. To objectively evaluate bowel cleanliness, we focused on artificial intelligence (AI)-based assessments. We aimed to generate a large segmentation dataset from CE images and verify its quality using a convolutional neural network (CNN)-based algorithm. *Materials and Methods*: Images were extracted and divided into 10 stages according to the clean regions in a CE video. Each image was classified into three classes (clean, dark, and floats/bubbles) or two classes (clean and non-clean). Using this semantic segmentation dataset, a CNN training was performed with 169 videos, and a clean region (visualization scale (VS)) formula was developed. Then, measuring mean intersection over union (mIoU), Dice index, and clean mucosal predictions were performed. The VS performance was tested using 10 videos. *Results*: A total of 10,033 frames of the semantic segmentation dataset were constructed from 179 patients. The 3-class and 2-class semantic segmentation’s testing performance was 0.7716 mIoU (range: 0.7031–0.8071), 0.8627 Dice index (range: 0.7846–0.8891), and 0.8927 mIoU (range: 0.8562–0.9330), 0.9457 Dice index (range: 0.9225–0.9654), respectively. In addition, the 3-class and 2-class clean mucosal prediction accuracy was 94.4% and 95.7%, respectively. The VS prediction performance for both 3-class and 2-class segmentation was almost identical to the ground truth. *Conclusions*: We established a semantic segmentation dataset spanning 10 stages uniformly from 179 patients. The prediction accuracy for clean mucosa was significantly high (above 94%). Our VS equation can approximately measure the region of clean mucosa. These results confirmed our dataset to be ideal for an accurate and quantitative assessment of AI-based bowel cleanliness.

## 1. Introduction

Since its introduction, capsule endoscopy (CE) [1] has been widely used to diagnose intestinal diseases. CE is minimally invasive and provides a physician with a view of the internal intestine for a diagnosis. A clean view of the intestine plays a crucial role in an accurate diagnosis. Unfortunately, one of the major issues, that is, the cleanliness of an intestine, does not always guarantee a clean view [2,3,4,5]. Traditional endoscopies such as duodenofibroscopy or ileocolonoscopy achieve a clean view with the help of irrigation or suction methods, while CE is passive and cannot be operated manually. For example, excessive bubbles, bile juice, ulcer debris, or bleeding can interfere with the mucosal vision in CE, and one may not be able to obtain an accurate diagnosis [2,3,4,5]. Therefore, assessing the visible region to distinguish between normal and abnormal mucosa is essential to a study’s qualitative verification [6]. To assess the visible mucosa region, bowel cleanliness score, which is usually obtained by a physician’s subjective reading, has been widely used. However, this operator-dependent score is subjective and lacks reproducibility [3,7,8]. Therefore, computer-aided mucosal vision assessment has received significant attention in recent years. For a brief review, Van et al. [9] provided objective measurements based on the color intensities of a tissue color bar. Similarly, Abou et al. [10] established a cleansing score, which is defined as the ratio of red to green pixels. Both these studies primarily focus on a color intensity-based approach, which is quite simple but unable to capture spatial information from an image; thus, it has a disadvantage in producing inaccurate cleansing scores. Nam et al. [11] introduced another approach: a learning-based method. They turned the cleansing score estimation into an image classification problem. They prepared CE images according to the size of the visualized mucosa, which were divided into five classes, and obtained a performance comparable to that of a physician’s direct examinations. However, to obtain an accurate quantitative visualization, five scales seem insufficient. Therefore, we created a large semantic segmentation dataset in this study for CE mucosal visual quantitative evaluation (a 10-visualization scale (VS) dataset). An overall flowchart of our study is depicted in Figure 1. Furthermore, we developed a VS formula from the semantic segmentation results.

## 2. Materials and Methods

Our primary endpoint is creating a CE semantic segmentation dataset in which the size distribution of the clean mucosal area exhibits as uniform as possible. In addition, our secondary outcome is verifying that our dataset is well constructed by training and testing CNN models.

### 2.1. Data Collection

Between January 2018 and June 2021, 179 capsule endoscopy studies (PillCam SB3, Medtronic, Minneapolis, MN, USA) performed in Pusan National University Yangsan Hospital (PNUYH) were collected and reviewed. CEs obtained from outside hospitals or incomplete study were excluded. We divided each image according to the size of the clean region, that is, stages 1–10 (from the largest clean region to the smallest one). Two gastroenterologists (Y.J. Lee, 13 years, and S.W. Moon, 5 years of experience in CE reading) generated sample images for each 10-VS stage, and five well-trained annotators, who have experience in more than 10,000 medical images annotation, selected 10,033 frames (approximately 1000 frames for each stage) from all CE images. After selecting the images, a data processing/refinement was performed using the following methods. This study was approved by the Institutional Review Board (IRB) of PNUYH (04-2021-049). As the study was retrospectively designed, the IRB approved the waiver of an individual’s informed consent.

### 2.2. Data Processing/Refinement

We categorized the pixel regions of a given CE image into three or two classes. All images were processed by five well-trained annotators, and the processed images were assessed, modified, and confirmed by two gastroenterologists (Y.J. Lee and S.W. Moon). The ground truth (GT) region of clean mucosa (GT VS) was calculated from the true semantic region.

3-class annotation: (1) floats/bubbles region where normal mucosa is not visible because of debris, blood, floats, bubbles, etc.; (2) dark region where there is an absence of light without disturbing the floats/bubbles; and (3) clean region where one can clearly observe the small intestinal mucosa (Figure 2a).

2-class annotation: (1) non-clean regions with the presence of floats/bubbles and dark regions and (2) clean regions where one can clearly observe the small intestinal mucosa. Namely, we merged floats/bubbles and dark regions into a non-clean region (Figure 2b).

### 2.3. Implementation Details

In order to verify the quality of our constructed dataset, we used three types of CNN architectures, i.e., DeepLab v3 [12], FCN [13], and U-Net [14], which are famous models designed for performing semantic segmentation task. Commonly, they roughly consist of encoder and decoder, in which a set of layers are stacked. In particular, DeepLab v3 has an atrous spatial pyramid pooling (ASPP) module in the encoder. We displayed the detailed configuration for each architecture in Table 1.

A raw sample from a CE video has a size of 576 × 576. We center-cropped a raw sample to remove unnecessary pixel regions, which usually include an image capture time and a PillCam logo. Consequently, we obtained samples with a size of 512 × 512. Moreover, we resized the center-cropped images to 256 × 256, 512 × 512, and 572 × 572 for DeepLab v3, FCN, U-Net, respectively (we used different resized size for better optimization). In addition, we used channel-wise mean and standard deviation (std) across the architectures to normalize the input with the following values: mean = (0.485, 0.456, 0.406) and std = (0.229, 0.224, 0.225) for the RGB channel. Apart from U-Net, the pretrained ResNet50 [15] and VGG16 [16] were adopted to build the backbone and encoder for DeepLab v3 and FCN, respectively. We used a batch size of 64, 100 epochs, polynomial learning rate scheduling, class balanced weights, and cross-entropy loss across the architectures.

### 2.4. Evaulation Metrics

To evaluate the semantic segmentation performance, we used the mean intersection over union (mIoU), which is equivalent to Jaccard index, and Dice index; both are defined as follows:$$\mathrm{mIoU}=\frac{1}{N}\overset{N}{\underset{k=1}{\sum }}\frac{\left|A_{k}\cap B_{k}\right|}{\left|A_{k}\cup B_{k}\right|},\;\;\;\mathrm{Dice}\;\mathrm{index}=\frac{1}{N}\overset{N}{\underset{k\;=1}{\sum }}\frac{2\left|A_{k}\cap B_{k}\right|}{\left|A_{k}\right|+\left|B_{k}\right|}$$
where N denotes the number of region classes, $A_{k}$ is the set of pixels belonging to the ground truth region for the k-th class, and $B_{k}$ is the set of pixels belonging to the predicted region for the k-th class. |·| is the cardinality of the set.

### 2.5. Visualization Scale

Recall that we are aiming to build a prediction algorithm that yields a VS based on semantic segmentation. Hence, we established a formula to appropriately represent a VS based on predicted regions as follows:(1)$$\mathrm{VS}=\frac{1}{M}\sum_{i=1}^{M}\frac{\left|C_{i}\right|}{\mathrm{size}\left(\mathrm{image}\right)}=\frac{1}{M}\sum_{i=1}^{M}\frac{\mathrm{size}\left(\mathrm{image}\right)-\left|D_{i}\cup F_{i}\right|}{\mathrm{size}\left(\mathrm{image}\right)},0\leq \mathrm{VS}\leq 1,$$
where M denotes the total number of frames from a patient, size(image) is the pixel size of a given image (= the center-cropped and resized image). $C_{i},{\;D}_{i}$, and $F_{i}$ denote the set of pixels belonging to clean, dark, and floats/bubbles regions for 3-class, respectively. Equation (1) describes the ratio of clean region to image size. For the 2-class case, the term $\left|D_{i}\cup F_{i}\right|$ is equivalent to the set of pixels belonging to a non-clean region. In addition, it is obvious that a patient without dark and floats/bubbles regions would result in a value of 1 on the VS, and vice versa.

## 3. Results

### 3.1. Data Details

Among the 179 CE studies (approximately over 8 million small bowel frames) from 179 patients, 10,033 frames of images were selected. In addition, 10,033 frames were split into 10 stages according to their clean region size. Figure 3 plots the class balance using the percentage of mean size for each class (floats/bubbles, dark, and clean) according to the stage number, and shows that the size of the clean region decreases linearly with the stage number. The mean age of patients was 46 ± 26 years (median: 52.2 years, range: 6.2–88.3 years), the mean body surface area was 1.6 ± 0.2 m^2^ (range: 0.7–2.3 m^2^), and 62% of the participants were male. A median of 35 frames (16, 65, IQR) and an average of four stages were extracted per person.

Table 2 shows a sample image and its corresponding ground truth (GT) image. In stage 1, neither floats/bubbles nor dark regions are present in an image; on the other hand, bubble and dark regions are present in the entire CE image in stage 10.

Overall, 7988 images from 169 patients were used for training, and 2045 samples from ten patients were used for testing. The statistics of the training and testing samples are listed in Table 2.

### 3.2. Experimental Results

#### 3.2.1. Overall Performance

As listed in Table 3, we measured overall 3-class performance for each architecture in terms of both mIoU and Dice index. All architectures showed semantic segmentation performance above 0.75 (mIoU) and 0.85 (Dice index). In addition, as listed in Table 4, overall performance of the 3-class and 2-class using DeepLab v3 was 0.7716 and 0.8972 by mIoU (Jaccard Index), 0.8627 and 0.9457 by Dice index, respectively.

#### 3.2.2. Detailed Results

The performances of VS prediction are listed in Table 4. We listed the mIoU and Dice index of ten tested patients along with their individual GT VS, our predicted VS for 2-class and 3-class, and example images (Table 4).

3-class performance: the overall 3-class performances were 0.7716 (ranging from 0.7031 to 0.8071) and 0.8627 (ranging from 0.7846 to 0.8891) in terms of mIoU and Dice index, respectively. The absolute difference between GT VS and measured VS for all patients was less than 0.033 (ranging from 0.0008 to 0.033). In addition, Patients 4 and 7, whose images had a cleaner view, had higher VS values than those of other patients.

2-class performance: the overall 2-class performances were 0.8972 (ranging from 0.8562 to 0.9330) and 0.9457 (ranging from 0.9225 to 0.9654) in terms of mIoU and Dice index, respectively. The absolute difference between GT VS and our VS for all patients ranges from 0.0047 to 0.0209.

Comparison of 3-class and 2-class: GT VS values obviously remains identical for both 3-class and 2-class, and our predicted VS exhibited nearly identical value for each patient across 3-class and 2-class. For all patients, 2-class exhibited better performance compared to 3-class in terms of both mIoU and Dice index. Patient 4 showed a huge ranking gap between 3-class and 2-class, i.e., jumping from 10th to 2nd. Patient 4 had relatively higher confusion between floats/bubbles region and dark region (Table 5), and this confusion absolutely disappeared in 2-class. For patient 3, the huge area of dark region was misclassified as the floats/bubbles region (Table 5); this confusion occurred when both floats/bubbles and dark region were present in the same area.

To provide more detailed results for a single patient (patient 5), we showed an example input image of the corresponding GT images and predicted images according to the stages, as displayed in Table 6. Notably, the overall shape of the predicted region was nearly identical to that of the GT region in both 3-class and 2-class classifications.

#### 3.2.3. Classification of Visible (Clean) Region

3-class performance: we achieved a 94.4% accuracy in clean region classification, as can be seen in Figure 4a, which shows the pixel-level confusion matrix. Apart from the pixel confusion in the dark and floats/bubbles regions (floats/bubbles → (dark, clean) with 9.8% probability, dark → (floats/bubbles, clean) with 14.2% probability), almost all true pixels for the clean region were correctly classified. This implies that our prediction algorithm can satisfactorily capture clean regions when compared to other regions.

2-class performance: we achieved a 95.7% accuracy for clean region classification as can be seen in Figure 4b. We conjecture this 1.3% (=95.7–94.4%) gap to be because of the different random weight initializations for DeepLab v3.

## 4. Discussion

We established a computer-aided VS based on semantic segmentation results using deep neural networks.

Semantic segmentation aims to cluster the pixels of an image belonging to the object of interest [17]. Because semantic segmentation results are used to realize a computer-aided VS, we used a deep learning-based semantic segmentation. Furthermore, we considered the most important reason for studies on semantic segmentation: it is consistent with a physician’s VS measurement. Although there are variations in VS measurements among individual physicians, they commonly regard the size of a clean region as crucial to estimating VS. Namely, assessing the size of a clean region could be a striking feature. Prior to our study, no such studies based on semantic segmentation have been reported. Our study is the first of its kind to apply semantic segmentation for building VS.

We describe our study’s main contributions, advantages, and limitations as follows.

### 4.1. Building a Semantic Segmentation CE Dataset

First, CE images were obtained from 179 patients. To improve the generalization of deep neural networks, we divided each image according to the size of clean region, that is, stage 1 to stage 10 (from the largest clean region to the smallest one). In addition, we confirmed a negatively linear relationship between the size of clean region and stage number. As for the number of segmentation classes, we assume that two distinct elements significantly hinder the visible region of mucosa. The first element is floats/bubbles, which mostly consist of small pieces of food debris and bubbles. The second element is the absence of light, which results in a dark region. Notably, the first element is strongly related to poor bowel preparation, whereas the second element is not. For the 3-class annotation, we first annotated the floats/bubbles and dark regions in a CE image. Once the annotation for these two regions was complete, a clean region was readily obtained (the remaining region). In addition, because the regions near the four corners of a sample are inevitably invisible because of the nature of the CE device, we also regarded them as clean regions for a simple annotation rule, and, by simply merging floats/bubbles region and dark region into a non-clean region, we could easily obtain the 2-class annotations. Following the abovementioned process, we eventually obtained 10,033 pairs of CE images and annotations.

### 4.2. Semantic Segmentation and VS Evaluation

We used three types of CNN architectures (DeepLab v3 [12], FCN [13], U-Net [14]) to verify that our dataset is well-constructed, and, we measured overall 3-class performance for each architecture in terms of both mIoU and Dice index. All architectures exhibited performance above 0.75 (mIoU) and 0.85 (Dice index). The results of DeepLab v3 showed the best performance compared with those of FCN and U-Net, which agrees with the results reported by Ahmed et al. [18]. Comprehensively, we could conclude that our dataset was well constructed. Since DeepLab v3 is the recent work compared with FCN and U-Net, we intensively conducted experiments and reported results based on DeepLab v3 (Table 4, Section 3.2.2 and Section 3.2.3). The 2-class exhibited better performance compared to 3-class due to two factors; (1) the absence of confusion between floats/bubbles and dark (Figure 4b), (2) the smaller number of classes. Furthermore, as mIoU and Dice index are positively correlated, one can easily notice that if one patient exhibits lower mIoU compared to the other patient, that is also true for Dice index (e.g., see patients 1 and 2). In addition, one can identify that the segmentation performance ranking gap between 3-class and 2-class for patient 4 is significantly different, i.e., jumping from 10th to 2nd, because patient 4 had relatively higher confusion between floats/bubbles and dark (Table 5), and this confusion absolutely disappeared in 2-class. There was another type of confusion, i.e., dark → floats/bubbles (Table 5). Commonly, this confusion occurred when both floats/bubbles and dark region were present in the same area.

Since VS calculation is strongly related to the size of the clean region, GT VS values obviously remain identical for both 3-class and 2-class. For similar reason, our predicted VS exhibited nearly identical value for each patient across both 3-class and 2-class.

### 4.3. Clean Region Prediction Performance

Using our established semantic segmentation dataset and DeepLab v3, we achieved 3-class and 2-class mIoUs of 0.7716 and 0.8972, Dice index of 0.8627 and 0.9457, respectively. In addition, based on the sizes of the regions, we established a VS formula in Equation (1). In the 3-class case, despite confusion between the dark and floats/bubbles regions, our VS was approximately consistent with GT VS. This is because an accurate clean region prediction plays an important role in vs. Equation (1), which also holds for the 2-class case. The 2-class segmentation performance differs from that of the 3-class segmentation, and all mIoUs and Dice indices for each patient and the overall mIoU and Dice index for the 2-class segmentation are higher than the 3-class segmentation. We conjecture this to be because of the smaller number of classes and the absence of any confusion between the floats/bubbles region and dark region when compared to the 3-class segmentation. Nevertheless, our VS consistently produced nearly identical predictions regardless of the number of classes (see Table 4). In other words, our semantic segmentation algorithm satisfactorily captured clean regions.

### 4.4. Limitations of Our Study

In this study, we simply ignored dark regions to produce VS. However, dark regions mostly occur when light from a capsule is unable to reach the mucosa. Even though the dark region appears to be invisible in the current frame, it may be visible in the subsequent frames when the capsule moves forward. Thus, a dark region is a strictly an ambiguous region, where one cannot decide on a given image. Hence, a more precise handling method of dark regions by considering the knowledge between consecutive frames, should be addressed. However, tackling this issue requires accurate dark region prediction performance, which was the limitation of our study, as there was relatively large confusion between floats/bubbles region and dark region. Therefore, inaccurately predicting dark regions was one of the limitations of our study.

## 5. Conclusions

To summarize, we established a large-scale dataset and developed a semantic segmentation algorithm for each image. We think our study will provide a cornerstone for accurately obtaining bowel cleanliness from video CE. We attempted to quantitatively evaluate bowel cleanliness based on AI, which was not possible before. As a result, we achieved high clean region prediction performance while exhibiting a fairly severe confusion between floats/bubbles region and dark region. In addition, although a dark region is invisible because of the absence of light, it may be potentially visible in the next frame; therefore, it was segmented separately from the invisible (i.e., non-clean) region. We believe this to be one of the clues for accurately obtaining bowel cleanliness from moving images by handling such dark regions as potentially visible regions in subsequent studies. The further exploration and obtaining accurate predictions for dark regions will be carried out in future work.

## Figures and Tables

**Figure 1 medicina-58-00397-f001:**
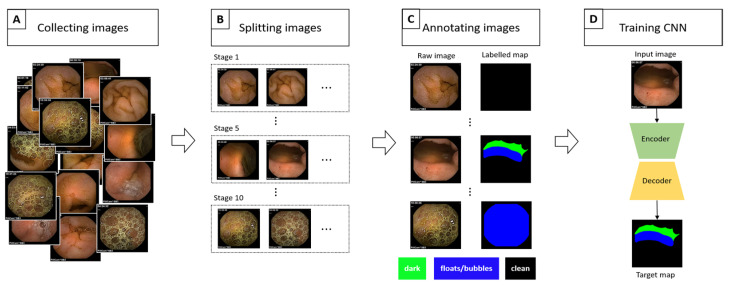
Overall flowchart of our semantic segmentation dataset construction. (**A**) We collected a set of raw images from CE. (**B**) Two gastroenterologists split the images into 10 stages according to the size of the clean area. (**C**) Each image was annotated by five well-trained annotators. (**D**) The quality of the constructed dataset was checked by training the CNN designed for semantic segmentation task.

**Figure 2 medicina-58-00397-f002:**
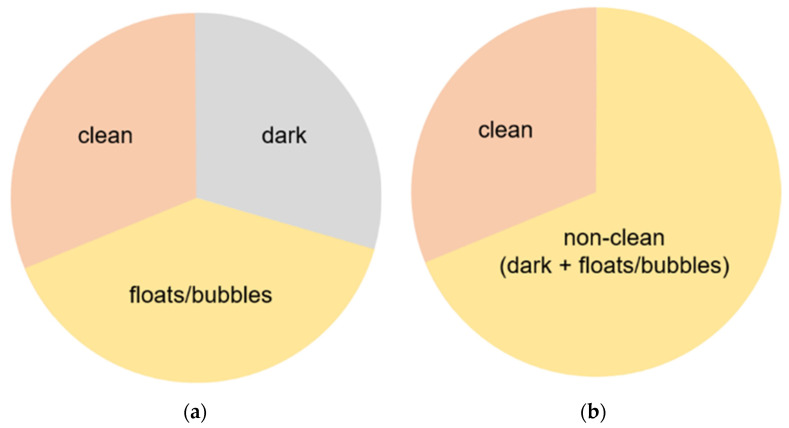
Schemes for each capsule endoscopy (CE) image class. Each color represents an individual region in a CE image. (**a**) 3-class: separate clean, dark, and floats/bubbles regions. (**b**) 2-class: Dark and floats/bubbles regions merged into a single class.

**Figure 3 medicina-58-00397-f003:**
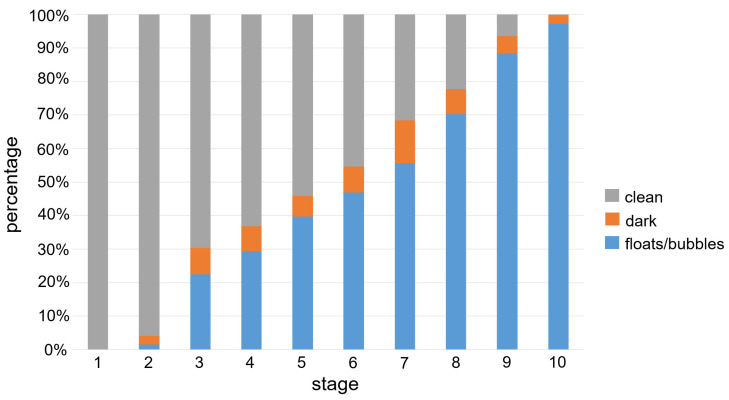
Class balance for each stage in our dataset. We calculate the percentage (vertical line) by averaging the ratio of the size of each class to image size.

**Figure 4 medicina-58-00397-f004:**
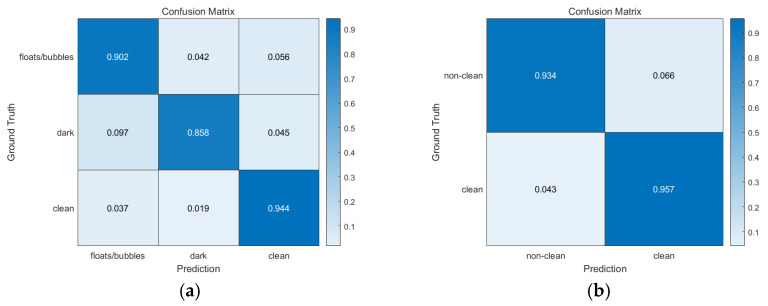
Pixel-level confusion matrix of our semantic segmentation algorithm based on DeepLab v3. (**a**) 3-class. Clean region exhibited better accuracy compared to other regions. (**b**) 2-class. Both clean and non-clean regions were fairly well classified correctly.

**Table 1 medicina-58-00397-t001:** Description of detailed configuration for each CNN architecture. [·] denotes a sequence of layers. 3×3×256 Conv (Rate 6) denotes a specific convolution layer; that is, kernel size: 3×3, channels: 256, dilatation rate: 6. Also, 2×2 UpConv consist of an up-sampling layer and 2×2 convolution.

Architecture	Encoder		Decoder
	Backbone	ASPP (Atrous Spatial Pyramid Pooling)	
DeepLab v3	ResNet 50	$\left[\begin{array}{c} 1\times 1\times 256\;\mathrm{Conv}\\ 3\times 3\times 256\;\mathrm{Conv}\;\left(\mathrm{Rate}\;6\right)\\ 3\times 3\times 256\;\mathrm{Conv}\;\left(\mathrm{Rate}\;12\right)\\ 3\times 3\times 256\;\mathrm{Conv}\;\left(\mathrm{Rate}\;18\right)\\ \mathrm{Global}\;\mathrm{Avg}\;\mathrm{Pool} \end{array}\right]$	$\left[\begin{array}{c} 3\times 3\times 256\;\mathrm{Conv}\\ \mathrm{Dropout}\\ 3\times 3\times 256\;\mathrm{Conv}\\ \mathrm{Dropout}\\ 1\times 1\;\mathrm{Conv} \end{array}\right]$
FCN	VGG16		$\left[\begin{array}{c} 3\times 3\times 512\;\mathrm{Conv}\\ 3\times 3\times 256\;\mathrm{Conv}\\ 3\times 3\times 128\;\mathrm{Conv}\\ 3\times 3\times 64\;\mathrm{Conv}\\ 3\times 3\times 32\;\mathrm{Conv}\\ 1\times 1\;\mathrm{Conv} \end{array}\right]$
U-Net	$\left[\begin{array}{c} 3\times 3\times 64\;\mathrm{Conv}\\ 3\times 3\times 64\;\mathrm{Conv}\\ 2\times 2\;\mathrm{Max}\;\mathrm{Pool}\\ \begin{array}{c} 3\times 3\times 128\;\mathrm{Conv}\\ 3\times 3\times 128\;\mathrm{Conv}\\ 2\times 2\;\mathrm{Max}\;\mathrm{Pool}\\ \begin{array}{c} 3\times 3\times 256\;\mathrm{Conv}\\ 3\times 3\times 256\;\mathrm{Conv}\\ 2\times 2\;\mathrm{Max}\;\mathrm{Pool}\\ \begin{array}{c} 3\times 3\times 512\;\mathrm{Conv}\\ 3\times 3\times 512\;\mathrm{Conv}\\ 2\times 2\;\mathrm{Max}\;\mathrm{Pool}\\ 3\times 3\times 1024\;\mathrm{Conv}\\ 3\times 3\times 1024\;\mathrm{Conv} \end{array} \end{array} \end{array} \end{array}\right]$.		$\left[\begin{array}{c} 2\times 2\;\mathrm{UpConv}\\ 3\times 3\times 512\;\mathrm{Conv}\\ 3\times 3\times 512\;\mathrm{Conv}\\ 2\times 2\;\mathrm{UpConv}\\ 3\times 3\times 256\;\mathrm{Conv}\\ 3\times 3\times 256\;\mathrm{Conv}\\ 2\times 2\;\mathrm{UpConv}\\ 3\times 3\times 128\;\mathrm{Conv}\\ 3\times 3\times 128\;\mathrm{Conv}\\ 2\times 2\;\mathrm{UpConv}\\ 3\times 3\times 64\;\mathrm{Conv}\\ 3\times 3\times 64\;\mathrm{Conv}\\ 1\times 1\;\mathrm{Conv} \end{array}\right]$

**Table 2 medicina-58-00397-t002:** Training samples and GT images for each visualization scale stage and its corresponding statistics. Images from capsule endoscopy (top row), 3-class ground truth region images (second row). Black, green, and blue denote clean, dark, floats/bubbles regions, respectively. 2-class ground truth region images (third row). Black and red denote clean and non-clean regions, respectively. The statistics of training/testing samples for each stage (bottom three rows). Measuring unit: images, GT: ground truth.

	Stage 1	Stage 2	Stage 3	Stage 4	Stage 5	Stage 6	Stage 7	Stage 8	Stage 9	Stage 10
Input Image	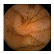	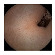	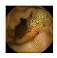	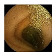	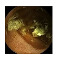	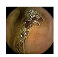	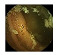	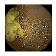	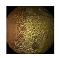	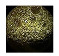
3-class	GT Image	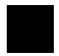	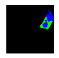	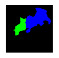	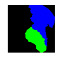	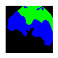	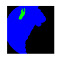	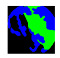	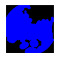	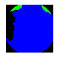	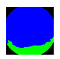
2-class	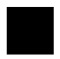	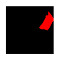	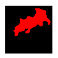	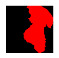	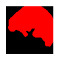	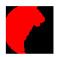	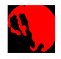	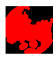	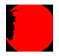	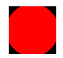
Training Examples Count	744	775	845	832	695	660	820	855	846	916
Testing Examples Count	256	234	173	179	286	327	186	147	154	103
Total Examples Count	1000	1009	1018	1011	981	987	1006	1002	1000	1019

**Table 3 medicina-58-00397-t003:** The 3-class semantic segmentation performance measured by mIoU and Dice index based on the results from DeepLab v3, FCN, and U-Net.

	mIoU (Jaccard Index)	Dice Index
DeepLab v3	0.7716	0.8627
FCN	0.7667	0.8602
U-Net	0.7594	0.8567

**Table 4 medicina-58-00397-t004:** Our semantic segmentation performance and its corresponding visualization scale (VS) on each test patient based on the results from DeepLab v3. Overall VS is not available since it is solely valid for an individual patient. (mIoU: mean intersection over union); GT VS (ground truth of visualization scale).

Patient Number	3-Class	2-Class	Example Image
mIoU (Jaccard Index)	Dice Index	GT VS	Predicted VS	mIoU (Jaccard Index)	DiceIndex	GT VS	Predicted VS
1	0.7743	0.8615	0.6355	0.6442	0.9070	0.9511	0.6355	0.6547	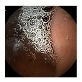
2	0.7964	0.8807	0.3598	0.3482	0.8988	0.9465	0.3598	0.3549	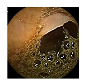
3	0.7097	0.8251	0.4471	0.4141	0.8562	0.9225	0.4471	0.4386	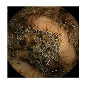
4	0.7031	0.7846	0.7498	0.7395	0.9260	0.9613	0.7498	0.7450	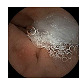
5	0.7303	0.8305	0.6069	0.5759	0.8718	0.9313	0.6069	0.5860	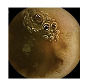
6	0.7655	0.8584	0.6743	0.6824	0.8783	0.9347	0.6743	0.6950	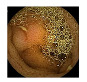
7	0.7608	0.8559	0.7374	0.7273	0.8826	0.9368	0.7374	0.7423	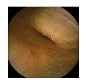
8	0.8054	0.8828	0.5007	0.4869	0.9330	0.9654	0.5007	0.4940	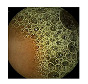
9	0.8071	0.8891	0.5190	0.5181	0.8928	0.9434	0.5190	0.5237	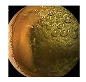
10	0.7990	0.8838	0.7179	0.7089	0.8924	0.9425	0.7179	0.7258	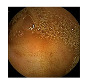
Overall	0.7716	0.8627	N/A	N/A	0.8972	0.9457	N/A	N/A	N/A

**Table 5 medicina-58-00397-t005:** Failure cases for Patient 3 and Patient 4 for the 3-class case. Confusion type (first row). Center-cropped images (second row). GT region annotated images (third row). Our prediction results (bottom row).

	Patient 3	Patient 4
Confusion Type	dark → floats/bubbles	floats/bubbles → dark
Example Image	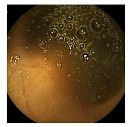	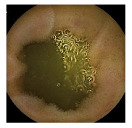
GT Image	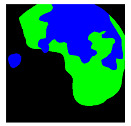	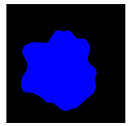
Prediction Image	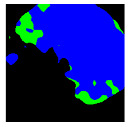	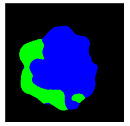

Black, green, and blue colors represent clean, dark, and floats/bubbles regions, respectively. GT: ground truth.

**Table 6 medicina-58-00397-t006:** Examples of test Patient 5. Input images (first row). Center-cropped images (second row). GT region annotated images and our prediction results for the 3-class case (third and fourth rows) and 2-class case (fifth and bottom rows).

.	Stage 1	Stage 2	Stage 3	Stage 4	Stage 5	Stage 6	Stage 7	Stage 8	Stage 9	Stage 10
Input Image	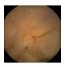	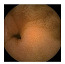	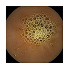	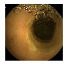	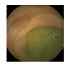	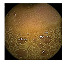	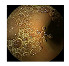	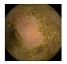	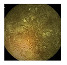	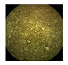
3-class	GT Image		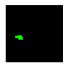	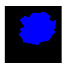	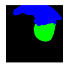	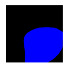	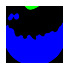	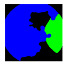	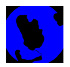	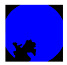	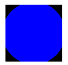
PredictionImage			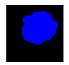	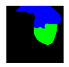	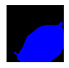	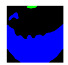	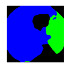	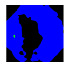	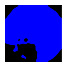	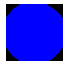
2-class	GT Image	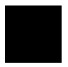		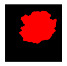	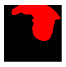	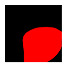	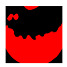	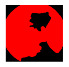	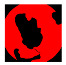	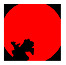	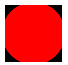
Prediction Image			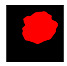	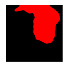	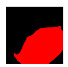	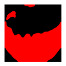	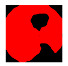	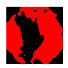	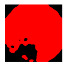	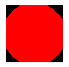

Black, green, blue, and red colors represent clean, dark, floats/bubbles, and non-clean regions, respectively. GT: ground truth.

## Data Availability

The data presented in this study are available on request from the corresponding author. The data are not publicly available because IRB approval does not involve data disclosure.

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
