# Peer review of "Semantic Segmentation Dataset for AI-Based Quantification of Clean Mucosa in Capsule Endoscopy"

_medicina, 2022, doi:10.3390/medicina58030397_

Round 1
Reviewer 1 Report
In this study, the authors proposed a method to evaluate the cleanliness of capsule endoscope images. They made a database containing endoscope images, their corresponding segmentation labels, and classification results, and performed the semantic segmentation task using Deep Lab. The results of the evaluation were acceptable, and the authors conclude that this study is effective in assessing the cleanliness of capsule endoscope images. I feel that this is a very interesting study and worthy of a special issue of this journal. I believe issues can be resolved:
- The title of this research paper includes "semantic segmentation dataset". It is suggested that the creation of the dataset is the main purpose of this research, and the availability of the data will contribute to the development of this research field. The availability of the database is required.
- The authors used images of four patients for the test, which is very small. More patient data needs to be used for accurate evaluation.
- The authors used Deep Lab as a method for semantic segmentation. There are no experimental results to support this choice, and it needs to be compared with other state-of-art models of segmentation, such as FCN, U-Net, and SegNet.
- It is better to evaluate the agreement of the segmented regions using Dice index and Jaccard index as well.
- It is recommended to provide a diagram of the outline of the research so that the process flow of the proposed method can be easily understood.
- A diagram of the network configuration should be added to the manuscript.
Author Response
Please see the the attachment

Reviewer 2 Report
The authors aimed to generate a large segmentation dataset from capsule endoscopy images and verify its quality using a convolutional neural network (CNN)-based algorithm. They concluded that the prediction accuracy for clean mucosa was significantly high (96%). Their visualization scale equation can approximately measure the region of clean mucosa.
This is a well written and designed manuscript. My concerns are as follows:
- References are inappropriate cited in the manuscript. Instead of the names of authors and year of publication, numbers in square brackets should be presented – Please see instructions to the authors and revise accordingly.
- Please present primary and secondary outcomes of the study in methodology.
- Please provide clear inclusion and exclusion criteria of the study.
- What was the level of experience of two gastroenterologists as well as annotators which were involved in this study. Please indicate in the methodology section.
- Tables 1 and 2 – What the numbers in both tables represent. Measuring units should be presented. Please revise.
- Limitations of the study should be clearly addressed at the end of the discussion section. Please revise.
- References should be cited as per journal style. Please see instructions to the authors and revise accordingly.
Round 2
Reviewer 1 Report
All issues have been resolved and will be considered accepted.